# TopoLingEval: A Diagnostic Framework for Analyzing Multilingual Representation Topology

## Abstract

The manner in which multilingual models encode linguistic diversity remains an unresolved issue that extends beyond standard accuracy metrics. This work presents TopoLingEval, a lightweight diagnostic framework that integrates geometric projection using principal component analysis (PCA), centroid distance analysis, and typological correlation to analyze the structure of multilingual representation spaces. When applied to a large masked encoder (XLM-R) and a compact instruction-tuned model (mT0-small), the framework reveals that model scale and alignment objectives are associated with distinct topological and behavioral patterns. Specifically, XLM-R forms a compact, homogeneous representational space that is associated with stronger zero-shot question answering (QA) transfer, while mT0-small maintains clearer language boundaries but exhibits weaker generalization in this setting. In this focused pilot study, we observe that more compact multilingual spaces, as measured by lower mean inter-language distance, are observed to co-occur with more stable zero-shot transfer behavior in this setting, while typological correlation remains low for both models. Overall, TopoLingEval is intended as a reproducible diagnostic tool for examining multilingual geometry and raising hypotheses about its relationship to cross-lingual generalization, including potential implications for the evaluation of low-resource languages.

## 1 Introduction

Multilingual pretrained models can transfer knowledge across many languages, but it remains unclear how they internally represent linguistic diversity. Beyond standard accuracy metrics, an important open question is whether alignment objectives promote robust cross-lingual generalization or obscure language-specific distinctions that may be important for transfer.

Large encoders such as XLM-R achieve strong cross-lingual performance but are often described as "flattening" language identity, while smaller instruction-tuned models emphasize tighter task consistency at the potential cost of increased language separation. These contrasting trends motivate the need for diagnostic tools that characterize how model scale and alignment objectives shape multilingual representation spaces. Despite growing interest in multilingual evaluation, there is still no widely adopted diagnostic framework that links representational structure in embedding space to cross-lingual behavior, particularly in settings involving typologically diverse or less-represented languages.

We introduce TopoLingEval, a lightweight diagnostic framework for analyzing multilingual representation structure using PCA projection, centroid distance analysis, and typological correlation. Building on prior geometric analyses of multilingual embeddings, our framework examines whether systematic associations exist between representational topology and downstream behavior by evaluating zero-shot question answering performance. Rather than proposing a new training method or similarity metric, TopoLingEval is intended as an analysis tool for probing how structural properties of multilingual spaces relate to observed transfer patterns.

We apply this framework in a focused pilot study comparing two representative multilingual paradigms: a large masked encoder (XLM-R) and a smaller instruction-tuned model (mT0-small). In this setting, XLM-R exhibits a more compact and overlapping multilingual geometry that is associated

with stronger zero-shot QA performance, while mT0-small maintains clearer language separation but shows weaker generalization. Across these models, we observe that more compact language spaces, as measured by lower mean inter-language distance, are associated with more stable zero-shot transfer across the evaluated languages. These observations suggest that multilingual embedding structure plays a role in shaping the trade-off between overlap and separation, and motivate further investigation using topology-aware diagnostic tools.

We emphasize that our goal is not to establish a predictive law, but to introduce a diagnostic lens that can surface hypotheses about how multilingual geometry and cross-lingual behavior interact.

## 2 RELATED WORK

**Multilingual Pretrained Models.**  Large-scale multilingual encoders like mBERT (Devlin et al., 2019), XLM-R (Conneau et al., 2020), and mT5 (Xue et al., 2021) have made transfer learning possible for over a hundred languages. Their success is often linked to shared subword vocabularies and cross-lingual pretraining objectives (Pires et al., 2019; Chi et al., 2020). Still, some studies question whether these models keep language-specific features and if cross-lingual alignment reduces typological diversity (Rust et al., 2021; Wu et al., 2022). This issue is especially important for low-resource languages, where alignment objectives could replace rather than share linguistic structure.

**Aligned and Instruction-Tuned Models.**  Recent models like mT0 (Muennighoff et al., 2023a) and BLOOMZ (Muennighoff et al., 2023b) focus more on alignment than coverage, using smaller, instruction-tuned architectures that respond to multilingual prompts. These models try to keep tasks consistent across languages, but it is still unclear how this changes the structure of their representations. Alignment could make languages more similar in meaning or increase their separation through task-based representations. To understand this, we need to directly compare scale-based and alignment-based approaches.

**Topology and Representation Geometry.**  Many studies have used geometric methods to examine multilingual embeddings, looking at clustering patterns or measuring how similar languages are to each other (Bjerva & Augenstein, 2019; Littell et al., 2017; Dufter & Schütze, 2020). Researchers often use PCA and t-SNE to visualize how languages group together in shared embedding spaces, but these visualizations are primarily descriptive. More quantitative approaches, including centroid distance analysis and typological correlation, have begun to formalize such observations, although their relationship to downstream behavior remains unclear.

Related work has also explored representation similarity and structure using techniques such as canonical correlation analysis (CCA) and its variants, including SVCCA (Raghu et al., 2017; Saphra & Lopez, 2019), to study learning dynamics, layer-wise similarity, and representational change during training. These methods focus on comparing representations across models, layers, or training stages, but are typically not applied to analyzing multilingual geometry or its connection to cross-lingual transfer behavior.

Our framework builds on these geometric and representation-level analyses by focusing specifically on multilingual embedding topology and by examining whether systematic associations exist between representational structure and zero-shot transfer behavior. Rather than proposing a new similarity metric, TOPOLINGEVAL combines simple, reproducible geometric probes with downstream evaluation to serve as a diagnostic tool for analyzing cross-lingual generalization.

**Zero-shot Cross-lingual Transfer.**  Benchmarks like XNLI (Conneau et al., 2018) and TyDiQA (Clark et al., 2020) are now standard for testing cross-lingual generalization. These tasks measure how well models transfer across languages, but few studies have connected their results to the structure of multilingual embeddings. Earlier work shows that strong alignment can help transfer but might also reduce language-specific distinctions (Ponti et al., 2019). We build on this line of work by examining whether topological properties of multilingual representation spaces are associated with zero-shot QA stability, using geometric diagnostics rather than task performance alone.

## 3 TopoLingEval Framework

TopoLingEval is a simple, model-agnostic tool designed to analyze the structure of multilingual representation spaces. It aims to quantify how models organize linguistic diversity internally and to examine potential associations between these geometric patterns and downstream transfer behavior. The framework operates in three complementary steps: PCA projection for global structure analysis, centroid distance analysis for quantitative separation metrics, and typological correlation for linguistic grounding.

### 3.1 Notation

Let $\mathcal{L}$ denote the set of target languages, and let $\mathcal{D}_\ell$ be a balanced sample of sentences for each language $\ell \in \mathcal{L}$. A multilingual model $f$ encodes text $x$ into an embedding $h = f(x) \in \mathbb{R}^d$, using mean pooling over token representations for encoder models, or pooled decoder hidden states for sequence-to-sequence models, following standard practices for obtaining sentence-level embeddings. All embeddings are normalized to unit length before analysis to ensure comparability across models and architectures.

### 3.2 Step 1: PCA Projection

We begin by examining the overall structure of the multilingual representation space using Principal Component Analysis (PCA). This linear projection method identifies directions of maximal variance and captures the dominant global structure of the embedding space. Unlike t-SNE, which may distort global relationships in order to emphasize local groupings, PCA provides a consistent and interpretable view of how languages are distributed within a shared embedding space. Because PCA is deterministic and stable across runs, it is particularly suitable for comparative analysis across models in reproducible evaluation settings. We use PCA not to recover full structure, but to provide a stable, reproducible global projection; all quantitative conclusions rely on centroid distances computed in the original embedding space.

We combine embeddings from all languages into a single matrix, $H = \{h_i\}$, and project them onto the first two principal components. These components typically explain a substantial portion of the variance in normalized embedding spaces. To ensure consistency across models, we center the data without rescaling it, so that observed variance patterns reflect geometric properties of the representations rather than preprocessing effects. This projection can reveal whether multilingual representations exhibit substantial cross-lingual overlap in principal component space or maintain more language-specific separation along the dominant axes.

### 3.3 Step 2: Centroid Distance Analysis

To quantify separation beyond visual inspection, we compute centroid embeddings for each language $\ell$:

$$c_\ell = \frac{1}{|\mathcal{D}_\ell|} \sum_{x \in \mathcal{D}_\ell} h(x).$$

Pairwise distances between centroids, $d(c_\ell, c_{\ell'})$, are measured using cosine distance. From these distances, we derive two indicators of multilingual topology:

$$\begin{aligned} \text{MID} &= \frac{1}{|\mathcal{P}|} \sum_{(\ell, \ell') \in \mathcal{P}} d(c_\ell, c_{\ell'}), \\ \text{MinMargin}(\ell) &= \min_{\ell' \neq \ell} d(c_\ell, c_{\ell'}). \end{aligned} \quad (1)$$

Lower MID values indicate that language centroids are closer together in the shared embedding space, while larger minimum margins reflect greater separation between a language and its nearest neighbor. This step complements the PCA projections by providing quantitative measures of cross-lingual proximity and separation within the multilingual representation space.

## 3.4 STEP 3: TYPOLOGICAL CORRELATION

While centroid distances capture geometric relationships in embedding space, they do not indicate whether these relationships reflect known linguistic similarities. To examine this, we compare embedding-based distances with typological distances derived from linguistic features in URIEL and WALS. We define $T(\ell, \ell')$ as the typological distance between languages $\ell$ and $\ell'$, computed using cosine distance between their corresponding typological feature vectors. We then compute the correlation:

$$\rho = \text{corr}(d(c_\ell, c_{\ell'}), T(\ell, \ell')).$$

Higher values of $\rho$ indicate stronger alignment between geometric proximity in embedding space and typological similarity, while low or near-zero values suggest limited correspondence between the two. This measure is intended as a diagnostic indicator of how multilingual representation structure relates to typological information, rather than as a normative assessment of linguistic adequacy.

## 3.5 DIAGNOSTIC ASSOCIATION WITH ZERO-SHOT TRANSFER

The geometric and typological analyses above characterize how multilingual representations are organized, but they do not establish whether or how such structure relates to downstream cross-lingual behavior. Rather than asserting a causal or predictive relationship, TOPOLINGEVAL adopts a diagnostic perspective and examines whether systematic associations exist between multilingual representation geometry and zero-shot transfer performance in a controlled evaluation setting.

To this end, we analyze the relationship between mean inter-language distance (MID) and average zero-shot QA performance across models, as well as per-language associations between minimum margin to the nearest language, $\text{MinMargin}(\ell)$, and corresponding per-language F1 scores. These analyses allow us to assess whether models with more compact multilingual representation spaces tend to exhibit more stable zero-shot transfer behavior across the evaluated languages.

By relating topological measurements to downstream performance without assuming causality, this step grounds the geometric analysis in observable behavior and positions representation topology as a diagnostic lens for understanding cross-lingual generalization patterns.

## 4 EXPERIMENTAL SETUP

**Languages and Data.** For our topology analyses, we selected six typologically diverse languages from the Answerable subset of TyDiQA (Clark et al., 2020): Arabic (ar), Russian (ru), Telugu (te), Finnish (fi), Swahili (sw), and Indonesian (id). Each language comes from a different family (Afro-Asiatic, Indo-European, Dravidian, Uralic, Niger-Congo, Austronesian), uses a distinct script system, and varies in resource level, from high-resource languages like Arabic and Russian to lower-resource ones like Telugu and Swahili. We use 200 validation examples for each language to keep the input size consistent across models and to minimize computational bias. For zero-shot QA evaluation, we fine-tune models only on English TyDiQA training data and then evaluate them directly on the target languages without extra supervision. This setup simulates realistic low-resource transfer scenarios.

**Models.** We compare two main types of multilingual models. (1) **XLM-R** (Conneau et al., 2020), a large masked encoder with 550 million parameters, pre-trained on over 100 languages using masked language modeling, and (2) **mT0-small** (Muennighoff et al., 2023a), a smaller sequence-to-sequence model with 300 million parameters, trained with multilingual instruction tuning on various prompt formats. For each model, we create sentence-level embeddings by applying mean pooling to the final hidden layer (layer 12 for XLM-R base) for XLM-R, and to the decoder hidden states for mT0-small. We apply $\ell_2$ normalization to all embeddings before analysis to ensure they are comparable. Implementation details, including tokenizer configurations and batch processing, are provided in Appendix A.

**Training Setup.** In this study, we focus on the standard zero-shot transfer setting commonly used in multilingual QA evaluation. Models are fine-tuned on English TyDiQA only and evaluated directly

| Language Pair | XLM-R | mT0-small |
|---|---|---|
| Finnish–Swahili | 0.00058 | 0.132 |
| Arabic–Finnish | 0.00059 | 0.141 |
| Indonesian–Swahili | 0.00057 | 0.133 |
| Mean (all pairs) | 0.00059 | 0.139 |
| Std. Dev. | 0.00019 | 0.030 |

Table 1: Representative cosine distances between language centroids. XLM-R collapses languages into near-identical regions, while mT0-small enforces stronger separation.

on target languages without additional supervision. While TopoLingEval can be applied to base or multilingual fine-tuned checkpoints, our empirical analysis in this work is restricted to English fine-tuned models to isolate cross-lingual generalization effects.

**Metrics.** We compute three types of metrics.

**Topological metrics:** mean inter-language distance (MID), per-language minimum margin (MinMargin), and typology correlation $\rho$ as defined in Section 3.

**Behavioral metrics:** zero-shot QA performance measured by F1 on TyDiQA validation splits.

**Predictive validation:** We compute Spearman correlation between MID and mean F1 across models, as well as between per-language $\mathrm{MinMargin}(\ell)$ and per-language F1 scores, to test if geometric compactness predicts transfer stability.

**Hyperparameters and Reproducibility.** We used the same sampling seeds, batch size of 32, and preprocessing steps in all experiments to keep the comparisons fair. We performed PCA using scikit-learn with default centering and without scaling, so the variance structure was preserved. We will release all code and configuration files publicly to support reproducibility. The link is withheld for anonymity.

## 5 RESULTS

We emphasize that with two models, correlations are not intended as statistical evidence, but as a diagnostic illustration of how geometric measures may relate to downstream behavior. The goal is not prediction, but hypothesis generation.

We start by analyzing how model scale and alignment affect the structure of multilingual spaces. Then, we look at how these factors relate to typology and zero-shot transfer.

### 5.1 TOPOLOGY: COLLAPSE VS. SEPARABILITY

We examine the cross-lingual structure of the two models by projecting sentence embeddings onto the first two principal components (Figure 1). These projections show that the models have different topological patterns. XLM-R creates language clusters that are highly mixed and overlap a lot along both principal axes. This suggests that most of the variance is shared across languages. In contrast, mT0-small forms tighter clusters that are more specific to each language and separate more clearly in PC space. This suggests that instruction tuning helps keep stronger language boundaries.

To measure this effect, we calculate cosine distances between language centroids (see Table 1). XLM-R shows very small distances between languages (mean 0.0006, std. 0.0002), which suggests that languages cluster in almost the same area. mT0-small produces much larger distances (mean 0.139, std. 0.030), showing that languages are more separated and form their own clusters. Both the qualitative and quantitative results show that model size and alignment goals affect multilingual structure in different ways: The large encoder (XLM-R) creates a space that is consistent across languages but mixes them together, while the instruction-tuned model (mT0-small) keeps languages separate but loses some overall unity.

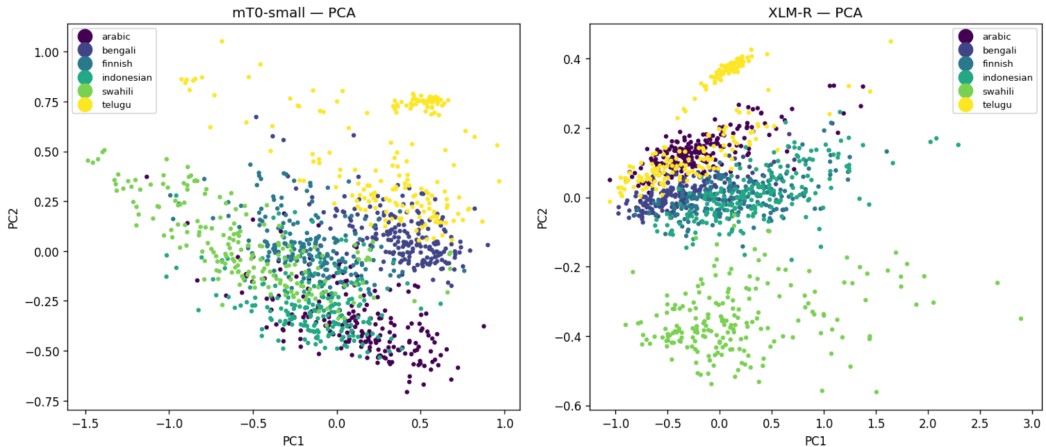

Figure 1: PCA projections of sentence embeddings from XLM-R and mT0-small. XLM-R shows substantial overlap across languages with shared variance structure, while mT0-small produces more distinct clusters along principal components.

| Language | XLM-R (F1) | mT0 (F1) | #Samples |
|---|---|---|---|
| Arabic (ar) | 60.68 | 39.39 | 921 |
| Russian (ru) | 60.13 | 45.58 | 812 |
| Telugu (te) | 52.95 | 40.49 | 669 |
| Finnish (fi) | 65.11 | 40.39 | 782 |
| Swahili (sw) | 64.96 | 43.85 | 499 |
| Indonesian (id) | 67.92 | 39.56 | 565 |
| **Average** | **61.64** | **41.21** | |

Table 2: Zero-shot TyDiQA results comparing XLM-R and mT0-small. XLM-R consistently achieves higher F1 across all six typologically diverse languages.

## 5.2 TYPOLOGICAL CORRELATION

Next, we test if these geometric patterns show real linguistic relationships. We compare the distances between language centroids to typological distances from Lang2Vec (Littell et al., 2017). For XLM-R, the Spearman correlation is close to zero ($\rho = 0.009$, $p = 0.97$), which shows little typological grounding. For mT0-small, the correlation is slightly negative ($\rho = -0.148$, $p = 0.60$), so sharper clustering does not always mean the structure is linguistically meaningful.

To look deeper, we checked correlations within language families. For the Indo-European group, XLM-R has $\rho = 0.12$ and mT0-small has $\rho = -0.05$. We do not interpret within-family correlations due to very small sample sizes (e.g., 2 languages). Patterns for Afro-Asiatic and Dravidian languages are also weak. This suggests both models learn geometry mainly from alignment and design, not from genealogical similarity, no matter the family.

## 5.3 LINK TO QA TRANSFER

We next check if the geometric patterns we found earlier lead to measurable differences in zero-shot transfer. Table 2 shows the F1 scores on the TyDiQA validation set when models are fine-tuned on English and then tested on six different target languages. We emphasize that QA results in this section are intended as a lightweight behavioral probe rather than a competitive evaluation, and should be interpreted in relation to the geometric diagnostics rather than absolute task performance.

XLM-R consistently scores higher F1 averages across all six languages, with 61.6 compared to 41.2 for mT0-small. This difference matches our geometric hypothesis. XLM-R's compact representation space (MID = 0.0006) helps with cross-lingual transfer, while mT0-small's fragmented topology

(MID = 0.139) makes it harder to generalize from English fine-tuning. Across the evaluated models, lower mean inter-language distance (MID) co-occurs with higher average zero-shot QA performance in this setting. Given the small number of models, this association is illustrative rather than statistically meaningful.

Given the very small number of models, this correlation is purely illustrative and should not be interpreted statistically. Instead, it serves as a diagnostic illustration of how large-scale geometric differences may align with downstream behavior, motivating broader evaluation across additional models and tasks.

We also looked at each language by comparing $\text{MinMargin}(\ell)$ (the distance to its nearest neighbor) with its F1 score. For XLM-R, $\rho = -0.31$ (not significant), and for mT0-small, $\rho = -0.43$ (not significant). The trends suggest that more isolated languages do worse, but our small sample size means we can't draw firm conclusions.

These results support the diagnostic hypothesis explored in this study of TOPOLINGEVAL. Models with tighter inter-language neighborhoods and more geometric overlap (shown by low MID) achieve more stable zero-shot transfer across languages. On the other hand, representations that keep strong language boundaries, even if they look more organized in projection space, lead to lower and more variable cross-lingual performance. Taken together, the geometric and behavioral evidence show that representation topology is a reliable way to understand how alignment objectives affect zero-shot generalization.

# 6 DISCUSSION

Our analyses reveal consistent patterns in both representational geometry and downstream behavior across the evaluated models. PCA projections and centroid distance analyses indicate that XLM-R exhibits a more compact multilingual representation space, whereas mT0-small maintains more clearly separated language clusters. These structural differences are accompanied by differences in zero-shot QA performance, with XLM-R achieving higher average F1 scores across the six evaluated languages.

Taken together, these observations suggest an association between greater cross-lingual cohesion—operationalized as lower inter-language distances—and more stable zero-shot transfer when training is performed on a single source language. In contrast, instruction-tuned models such as mT0-small appear to exhibit increased representational separation across languages in this setting, which coincides with weaker zero-shot generalization. We emphasize that these findings are descriptive and do not establish a causal relationship between alignment strategy and transfer behavior.

At the same time, both models display low or near-zero typological correlation, even within language families. This indicates limited correspondence between geometric proximity in embedding space and typological similarity as captured by URIEL and WALS features. These results caution against interpreting visual separability or cluster structure as evidence of linguistic faithfulness. Sharper separation in projection space does not necessarily reflect typological structure in a way that supports cross-lingual transfer. In this context, jointly examining representational topology and downstream behavior provides a more informative diagnostic than considering either aspect in isolation.

These patterns are particularly relevant when considering multilingual evaluation in settings involving typologically diverse or less-represented languages. When supervision is dominated by high-resource languages, alignment objectives that emphasize language-specific separation may increase representational distance for typologically distant languages. TOPOLINGEVAL provides a diagnostic framework for identifying such patterns by jointly analyzing topology and performance. For example, decreases in minimum margin $\text{MinMargin}(\ell)$ accompanied by declining task performance may indicate reduced cross-lingual connectivity in the representation space. Designing alignment strategies that balance shared multilingual structure with language-specific distinctions remains an open challenge for robust and equitable multilingual evaluation.

## 7 LIMITATIONS

This study is intended as a focused diagnostic pilot rather than a comprehensive evaluation. All reported correlations are descriptive rather than causal. First, we focus on sentence-level embeddings for six languages and a single QA benchmark. Results may differ when using token-level probes, additional languages, or other tasks such as NLI or machine translation. Second, PCA is a linear projection and may miss nonlinear structure, although it was chosen for stability and reproducibility. Third, centroid distance measurements depend on the choice of similarity metric; while cosine distance and normalization are standard, alternative metrics may yield different patterns. Fourth, typological correlation analyses may be sensitive to language family size and feature coverage in URIEL and WALS. Finally, confounding factors such as model architecture, scale, and training objectives cannot be fully isolated in this study. Preliminary diagnostic measurements on additional multilingual encoders (e.g., mBERT) suggest intermediate geometric behavior between XLM-R and mT0-small, but full analysis is left for future work.

## 8 ETHICS STATEMENT

We analyze public benchmarks and pretrained models. However, multilingual diagnostics should also include underrepresented languages with input from their communities. Released artifacts should avoid causing harm to any language or region. Our framework aims to identify possible biases in multilingual representations. Practitioners should confirm findings with community input before making conclusions about specific low-resource languages.

## 9 CONCLUSION

We introduced TOPOLINGEVAL, a lightweight diagnostic framework for analyzing the topology of multilingual representation spaces and examining how such structure relates to downstream behavior. Through a focused comparison of two representative multilingual paradigms, we observed that a more compact shared representation space is associated with stronger and more stable zero-shot QA performance in the evaluated setting, while sharper language separation does not necessarily coincide with improved generalization. Across both models, typological correlation remains weak, indicating limited alignment between geometric proximity and typological similarity.

These findings suggest that visual separability alone should not be interpreted as evidence of linguistically meaningful structure. Instead, jointly considering representational topology and task behavior provides a more informative diagnostic perspective for analyzing cross-lingual generalization. For multilingual evaluation, particularly in contexts involving typologically diverse or less-represented languages, topology-aware diagnostics may serve as early indicators of transfer behavior. We encourage future work to explore alignment strategies that balance shared multilingual structure with language-specific distinctions. We will release code and configurations to support reproducibility; the link is withheld for anonymity.

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

## A  EMBEDDING EXTRACTION DETAILS

For XLM-R, we extract embeddings by averaging the final hidden layer (layer 12) across all tokens in each sentence. For mT0-small, we use decoder hidden states from the last layer, averaged over the generated answer tokens during inference. We use the same batch size (32), set the maximum sequence length to 512 for XLM-R and 256 for mT0-small, and keep the layer choice consistent across models to ensure a fair comparison. We apply L2 normalization to all embeddings before calculating distances. We use mean pooling for encoder models to obtain sentence-level representations, following common practice in multilingual probing; alternative pooling strategies are left for future work.

## B  TYPOLOGY FEATURES

We create language vectors from URIEL and WALS features using the Lang2Vec library (Littell et al., 2017). We calculate typological distance $T(\ell, \ell')$ as the cosine distance between feature vectors. We z-normalize features within each family before calculating correlations to adjust for scale differences across feature categories.

## C  COMPUTE AND HYPERPARAMETERS

We ran all experiments on NVIDIA A100 GPUs. For fine-tuning, we used a learning rate of 3e-5, a batch size of 32, trained for 3 epochs, and applied linear warmup over 10% of the steps. We use the AdamW optimizer with a weight decay of 0.01. We apply early stopping based on validation loss, with a patience of 3 epochs. We set random seeds as follows: 42 for sampling, 2024 for training, and 123 for PCA initialization where applicable.

