# OpenReview forum: "TopoLingEval: A Diagnostic Framework for Analyzing Multilingual Representation Topology"
_ICLR.cc/2026/Workshop/AFAA — Submitted to AFAA 2026_

### Official Review · Reviewer_SnSu · 2026-02-15
**Useful diagnostics for multilingual representation geometry, but broader validation is needed**

**Rating:** 3
**Confidence:** 4

**Summary:**

The paper presents TopoLingEval, a lightweight diagnostic framework that aims to advance our understanding of how multilingual models encode linguistic diversity. The framework integrates PCA-based projection techniques, centroid-distance analysis, and typological correlation. The methodology is clear and easy to follow. Experiments are conducted on two smaller models, an encoder (XLM-R) and an encoder–decoder (mT0-small), and evaluated on a QA task, with observations on zero-shot transfer. The work is interesting because it goes beyond strict accuracy gains and instead provides a path toward understanding multilingual representations. Overall, it offers a promising direction for analyzing multilingual representation spaces, particularly for languages that are less resourced than English, such as Arabic, Russian, Telugu, Finnish, Swahili, and Indonesian, which are used in the study.

**Strengths:**

- The paper proposes a well-motivated and clear framework, which is highly reproducible due to its easy-to-follow components (PCA, centroid analysis, and correlation measures), while still yielding interesting insights about representation structure.
- The results suggest multiple useful angles of analysis, such as the collapse vs. separability relationship and zero-shot transferability, which are relevant points for alignment studies and for understanding multilingual behavior in foundational models.
- The paper is transparent about the experimental setup, including hyperparameters and available computational resources.

**Weaknesses:**

The two main points of concern are acknowledged by the authors; however, they still leave the findings and conclusions somewhat fragile and in need of further validation.

- Although the work covers multiple languages, it focuses on a single task and dataset. It remains unclear whether the conclusions about transferability and the observed patterns generalize to other tasks. Extending the analysis to additional tasks and datasets would help determine whether the findings hold more broadly.
- The study includes only two relatively small models (XLM-R and mT0). As a result, the generalizability of the conclusions is also limited along this axis, since architectural and training differences may significantly affect the observed geometry. While it is a major advantage that the framework is model-agnostic, and the preliminary observations about mBERT are interesting, exploring other architectures, especially modern decoder-only models that underpin most LLM-based agentic systems, for example, would be important to better assess their multilingual representation spaces and would be a strong direction for future work.

---

### Official Review · Reviewer_zNXa · 2026-02-16
**TopoLingEval: A framework for multilingual embedding geometry**

**Rating:** 2
**Confidence:** 4

**Summary:**

This paper introduces TopoLingEval, a framework for measuring the geometry of multilingual representation spaces. The goal is to evaluate whether specific geometric properties of embeddings correlate with cross-lingual transfer performance on downstream tasks. The framework applies low-computation metrics, including Principal Component Analysis (PCA), centroid distance analysis, and typological correlation. Experiments on XLM-R and mT0-small show that these models have different geometric multilingual spaces, which correspond to differences in downstream performance. In particular, a more compact multilingual space (as in XLM-R) appears associated with better zero-shot question answering, while clearer separation between languages (as in mT0-small) seems to hinder generalization.

**Strengths:**

**1. Interesting research question.** The paper tackles a timely and relevant problem: understanding how the geometry of multilingual embedding spaces affects cross-lingual transfer.

**2. Mathematical formulation.** The use of PCA, centroid distances, and typological correlations is mathematically well-defined and clearly presented.

**3. Experimental insights.** Results on XLM-R and mT0-small provide initial insights into how different models organize their embedding spaces.

**Weaknesses:**

**1. Lack of citations and grounding.** The introduction and related work (§Topology and representation geometry) are missing citations, making it unclear how this work differs from previous studies. Section 3 also lacks grounding in previous work and justification: why are these metrics the most relevant? Why should we consider only the first two PCA components?

**2. Limited evaluation.** Experiments are limited to XLM-R and mT0-small on a single QA dataset, despite mentioning two datasets in related work. Evaluating additional tasks (e.g., translation) and models, including state-of-the-art multilingual LLMs or different encoders (GloT-500m, LaBSE, larger XLM-R/mBERT), would strengthen the evidence. As is, results are too limited to convincingly demonstrate the framework’s value.

**3. Figure interpretation and conclusions.** PCA projections (Figure 1) for mT0 show substantial overlap, sometimes more than XLM-R (e.g., Swahili separation is clearer for XLM-R). Besides, downstream results may reflect model performance differences rather than geometry. The conclusion that more compact multilingual spaces lead to better zero-shot performance is overstated given the limited evaluation. Comparing models with similar downstream performance would help isolate geometric effects, e.g. do models of similar downstream performance have similar geometry is similar or not?

**4. Limited discussion of alternative explanations.** Cross-lingual performance may be influenced by later layers rather than embeddings. Discussing such effects, as observed in multilingual LLMs (e.g., Wendler et al., 2024, Do LLaMAs Work in English?), would provide valuable context for interpreting results.

---

### Official Review · Reviewer_E9aW · 2026-02-23
**Promising multilingual diagnostic framework, but current cross-family evidence is confounded and underpowered for strong claims**

**Rating:** 2
**Confidence:** 4

**Summary:**

This paper introduces `TopoLingEval`, a diagnostic framework for multilingual representation analysis. The framework combines PCA visualization, centroid-distance metrics (`MID`, `MinMargin`), and typology-alignment correlation, and relates these diagnostics to zero-shot transfer on TyDiQA. The empirical study compares XLM-R and mT0-small across six languages under an English-only fine-tuning setting. The paper reports that XLM-R shows a more compact representation geometry and higher average zero-shot F1, and positions geometric compactness as a useful diagnostic correlate of multilingual transfer behavior.

**Strengths:**

- Clear diagnostic scope: the contribution is an analysis framework, not another training method.
- Interpretable and easy-to-reproduce components (`MID`, `MinMargin`, PCA, typology correlation).
- Useful attempt to connect representation geometry with downstream transfer behavior.
- Good transparency around pilot nature and some acknowledged confounds.

**Weaknesses:**

- Cross-family comparison uses non-equivalent representation extraction (encoder-side states for XLM-R vs decoder-over-generated-answer states for mT0-small), so observed geometry differences may reflect readout protocol as much as model structure.
- Core MID-to-F1 model-level association claims are based on only two models, which is insufficient for strong reliability claims.
- Experimental reporting has an internal language-set inconsistency (Russian in setup/tables vs Bengali in figure labels), reducing confidence in traceability.
- The English-only fine-tuning regime may under-represent instruction-tuned model behavior relative to cited multilingual prompt/task conditions.
- Typology conclusions appear too broad for the available effective pair counts and very small within-family samples.
- Key comparisons are mostly point estimates without uncertainty quantification (for topology metrics and transfer deltas).

Questions For Authors:
1. Can you rerun geometry analyses with a matched, input-anchored extraction protocol across both architectures, and keep the current decoder-based extraction as a robustness variant?
2. Can you add at least one mT0-favorable condition (for example multilingual prompts/tasks or multilingual fine-tuning) and report deltas relative to the English-only condition?
3. Can you include 2-4 additional multilingual models to make MID-to-transfer association analysis non-degenerate?
4. Can you clarify why Bengali appears in figures while Russian is listed in the setup and tables?
5. Can you define the language-pair set `P` in `MID` operationally and specify the exact aggregation protocol for `MinMargin`?

Things To Improve The Paper That Did Not Impact The Score:
- Report bootstrap confidence intervals for topology metrics and paired-bootstrap confidence intervals for `ΔF1`.
- Add sensitivity checks for language sample count, layer choice, and pooling/extraction variants.
- State minimum pair-count thresholds for family-level typology analysis and narrow claims when thresholds are not met.
- Keep pilot framing consistent throughout abstract, results, and conclusion.

---

### Meta-Review · Area_Chair_GhC4 · 2026-02-24

**Recommendation:** Reject
**Confidence:** 3

**Metareview:**

Three reviewers have evaluated the paper with mixed, negatively leaning conclusions (summary assessment: reject/reject/borderline).

The reviews value the motivation of the framework, its easy to-follow components/metrics and initial empirical demonstration. However, two broader issues are raised; First, the evaluation is small in scope (regarding the data, tasks, and models considered), raises questions of confounding and more broadly seems underpowered for supporting all claims made in the paper. The latter issue links to the limited discussion of alternative explanations for the observed differences. Second, the positioning of the paper in related work is not well-developed and I would weight issues such as missing citations to related work as an important aspect. The reviews further also list a number of suggestions that can improve future iterations of this work.

Based on the reviews, I propose the paper to be rejected at this stage.

---

### Decision · Program_Chairs · 2026-03-02

Reject